# Initiation of HIV-1 Gag lattice assembly is required for recognition of the viral genome packaging signal

Xiao Lei[1], Daniel Gonçalves-Carneiro[1], Trinity M Zang[1,2], Paul D Bieniasz[1,2]*

[1]Laboratory of Retrovirology, Rockefeller University, New York, United States; [2]Howard Hughes Medical Institute, The Rockefeller University, New York, New York, United States

**Abstract** The encapsidation of HIV-1 gRNA into virions is enabled by the binding of the nucleo-capsid (NC) domain of the HIV-1 Gag polyprotein to the structured viral RNA packaging signal ($\Psi$) at the 5' end of the viral genome. However, the subcellular location and oligomeric status of Gag during the initial Gag-$\Psi$ encounter remain uncertain. Domains other than NC, such as capsid (CA), may therefore indirectly affect RNA recognition. To investigate the contribution of Gag domains to $\Psi$ recognition in a cellular environment, we performed protein-protein crosslinking and protein-RNA crosslinking immunoprecipitation coupled with sequencing (CLIP-seq) experiments. We demonstrate that NC alone does not bind specifically to $\Psi$ in living cells, whereas full-length Gag and a CANC subdomain bind to $\Psi$ with high specificity. Perturbation of the $\Psi$ RNA structure or NC zinc fingers affected CANC:$\Psi$ binding specificity. Notably, CANC variants with substitutions that disrupt CA:CA dimer, trimer, or hexamer interfaces in the immature Gag lattice also affected RNA binding, and mutants that were unable to assemble a nascent Gag lattice were unable to specifically bind to $\Psi$. Artificially multimerized NC domains did not specifically bind $\Psi$. CA variants with substitutions in inositol phosphate coordinating residues that prevent CA hexamerization were also deficient in $\Psi$ binding and second-site revertant mutants that restored CA assembly also restored specific binding to $\Psi$. Overall, these data indicate that the correct assembly of a nascent immature CA lattice is required for the specific interaction between Gag and $\Psi$ in cells.

*For correspondence:
pbieniasz@rockefeller.edu

**Competing interest:** The authors declare that no competing interests exist.

## Editor's evaluation

This work presents valuable findings that advance our understanding of the roles of the CA domain in specific binding of HIV-1 Gag to the viral genomic RNA. The compelling evidence obtained using the modified CLIP-seq and chemical crosslinking approaches support the authors' conclusion that the initial Gag lattice formation mediated by CA is essential for Gag recognition of the 5' $\Psi$ sequence. This work will be of interest to virologists working on gRNA packaging of not only HIV-1 but also other RNA viruses.

## Introduction

The generation of infectious HIV-1 virions depends on the encapsidation of two copies of HIV-1 gRNA into virions. HIV-1 gRNA is present among a vast excess of host RNAs in the cytosol of infected cells, yet it is the dominant RNA species in virions (*Rulli et al., 2007*). The specific encapsidation of HIV-1 gRNA into virions is attributed to the specific interaction between HIV-1 Gag and the viral packaging signal '$\Psi$' (*Aldovini and Young, 1990*; *Berkowitz et al., 1996*; *Clavel and Orenstein, 1990*; *Clever et al., 1995*; *Clever et al., 2000*; *Darlix et al., 1990*; *D'Souza and Summers, 2005*; *Lever et al.,*

*1989*). The Gag protein consists of three major domains: matrix (MA), capsid (CA), and nucleocapsid (NC). The NC domain binds directly to RNA, and point mutations or deletions disrupting the zinc fingers of NC interfere with the specific packaging of gRNA (*Aldovini and Young, 1990*; *Gorelick et al., 1990*). Nevertheless, since the NC domain is present in the context of Gag precursor when initiating interactions with Ψ, other domains of Gag may also contribute to Ψ binding specificity. Indeed, RNase protection assays, yeast-three hybrid assays, and in vitro binding assays suggest that full-length Gag or CANC has different RNA binding specificity compared to NC, and some studies indicate binding to Ψ with higher specificity than NC (*Bacharach and Goff, 1998*; *Damgaard et al., 1998*; *Guo et al., 2020*; *Kroupa et al., 2020*; *Webb et al., 2013*). Moreover, Gag mutants with CA mutations exhibited reduced gRNA selectivity in an in vitro reconstituted HIV-1 Ψ packaging system (*Carlson et al., 2016*), while Gag mutants with CA assembly deficits cannot initiate gRNA packaging (*Duchon et al., 2021*; *Kutluay et al., 2014*; *Kutluay and Bieniasz, 2010*).

Nevertheless, there remains significant uncertainty as to the location, stoichiometry, and the overall nature of the protein RNA complex that initiates HIV-1 assembly. To probe the potential role of CA and early assembly events in the specific interaction between Gag and Ψ in biologically relevant settings, we performed crosslinking immunoprecipitation (CLIP) experiments to determine whether full-length Gag, a CANC subdomain, or NC alone could specifically bind to Ψ within HIV-1 gRNA in cells. We found that full-length Gag and CANC both bind to Ψ with high specificity in the cytosol, whereas an isolated NC domain does not. Through further studies of CA mutants, we demonstrate that manipulations which perturb any of the CA:CA interfaces required for the formation of an immature Gag lattice also perturb the specific interaction between Gag/CANC and Ψ. Our findings suggest that assembly of a nascent immature Gag lattice is required for the initial recognition of Ψ and the initiation of viral RNA packaging in the infected cell cytoplasm.

## Results
### CLIP method for assessing Gag-RNA interaction in cells

We used a modified CLIP method based on published protocols (*Kutluay et al., 2014*; *Shema Mugisha et al., 2020*) to measure HIV-1 Gag-RNA binding interactions in living cells. In this CLIP procedure, an infrared-dye-conjugated 3' adaptor (*Zarnegar et al., 2016*) is used in place of a conventional radioactive labeled 3' adaptor. Protein-RNA crosslinking is driven by metabolic 4-thiouridine incorporation into target RNA (*Hafner et al., 2010*), and the labeled adaptor is ligated to protein/RNA crosslinked species directly on antibody-conjugated Dynabeads (*Figure 1—figure supplement 1*). These methodological adjustments cut the experimental time by half without loss of sensitivity, enabling higher throughput in CLIP assays.

For CLIP experiments, we used derivatives of an HIV-1 NL4-3 based proviral construct, referred to as HIV-1 NL4-3 (MA-3xHA/PR⁻) that carries a 3xHA epitope tag, between the MA and CA domains of Gag and encodes an inactivated protease (*Kutluay et al., 2014*; *Figure 1A*). As we intended to study the initial Gag:gRNA interaction in the cytosol and MA is not involved in this process (*Bou-Nader et al., 2021*; *Kutluay et al., 2014*), we also generated a proviral construct (referred to as pCANC) in which the start codon of MA (ATG) was mutated to abolish MA expression; instead, an N-terminally 3xHA CANC Gag fragment, along with the C-terminal SP2-p6 extension is expressed (*Figure 1A*). Crucially, the RNA binding profile of the CANC protein to the HIV-1 genome was similar to that of the full-length Gag protein, with a marginally higher fraction of reads mapping to Ψ. We speculate that this apparently enhanced preference of CANC for Ψ is because the membrane bound Gag, that exhibits more promiscuous binding to the viral genome (*Kutluay et al., 2014*), is not present when CANC is used.

Both the full-length Gag protein and CANC exhibited characteristic binding to Ψ involving three 'peaks' of read intensity corresponding to U5, SL1, and SL3/4 that are brought into proximity in the folded minimal packaging structure (*Bieniasz and Telesnitsky, 2018*; *Lu et al., 2011*; *Figure 1B and C*). To investigate the effects of RNA perturbation on CANC recognition, we prepared two Ψ mutants. In one mutant, the GC-rich 'kissing' loop (GCGCGC) in the dimerization initiation site (DIS) was substituted with a GAGA tetraloop (CANC DIS-GAGA) to prevent RNA dimerization (*Lu et al., 2011*). In a second mutant, four bases 'GGAA' were appended at the 5' end of the gRNA, immediately 5' to three guanosines at the transcription start site of HIV-1 viral genome (CANC InsGGAA). These nucleotides,

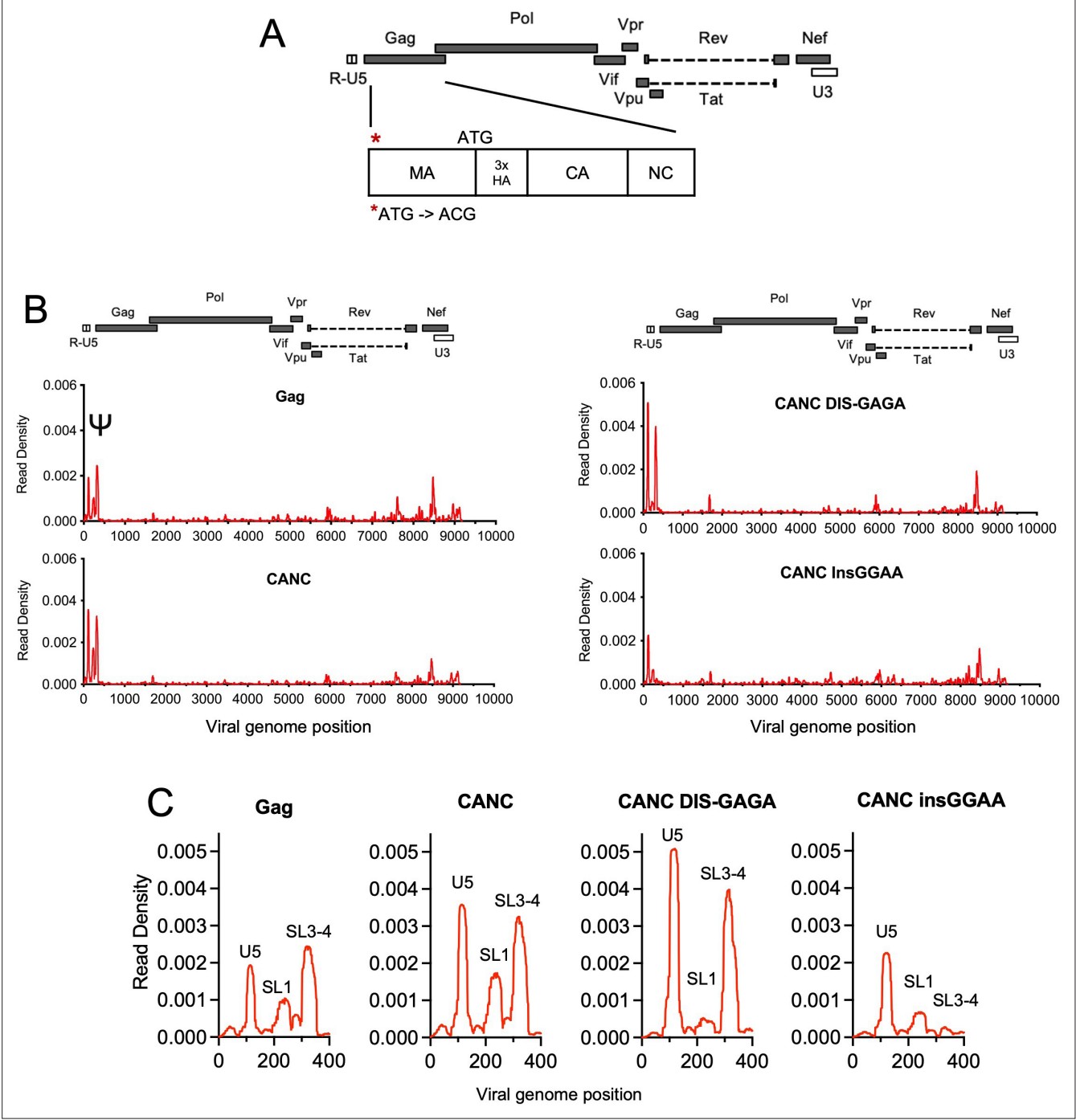

**Figure 1.** Crosslinking immunoprecipitation (CLIP) method validation and effects of Ψ perturbations on Gag recognition. (**A**) Schematic representation of the pCANC construct. Asterisk indicates the mutation introduced at the matrix (MA) start codon (ATG to ACG). (**B**) Read density distribution on viral RNA from CLIP experiments in which constructs encoding Gag, CANC, or CANC with mutant Ψ elements (CANC DIS-GAGA and CANC InsGGAA) were used. The y-axis represents the decimal fraction of all reads that mapped to the viral genome in which a given nucleotide was present. The x-axis indicates the nucleotide position on the viral genome. A colinear schematic HIV-1 genome is presented above each set of charts. (**C**) Expanded view of read densities for the 400 nucleotides at the 5' end of the viral genome. For (**B**) and (**C**), CLIP assays for each construct were repeated in at least two independent experiments, and the average read density is plotted.

The online version of this article includes the following source data and figure supplement(s) for figure 1:

**Source data 1.** Read density data for *Figure 1B* and *Figure 1C*.

**Figure supplement 1.** Crosslinking immunoprecipitation (CLIP) procedure.

along with the 5' cap affect the structure of the entire 5' leader (*Ding et al., 2021*). While CANC bound to the unmanipulated Ψ sequence with high specificity, the CANC InsGGAA Ψ mutant was bound comparatively poorly by CANC, with failure of the 3' portion of Ψ to bind CANC (*Figure 1B and C*). This finding is consistent with previous studies showing that the exposure of the 5'-cap of gRNA to the translational machinery due to 'GGAA' insertion inhibits gRNA packaging (*Ding et al., 2021*). The CANC DIS-GAGA Ψ mutant, which presumably favors the retention of a monomeric RNA, was bound by CANC specifically within the Ψ element, but with the exclusion of the SL1 stem-loop (*Figure 1B and C*). This finding is consistent with previous findings that other Ψ elements are retained in the DIS-GAGA structure as in native dimer (*Keane et al., 2015*) and suggests that the dimerized RNA elements are part of the structure recognized by Gag. Overall, these results validate the utility of the CLIP method used herein and are consistent with the finding that the tertiary, and to some extent the quaternary, structure of the Ψ element is important for accurate recognition by Gag.

## HIV-1 CA is required for specific binding to Ψ

We next compared the ability of various manipulated CANC proteins to bind to the gRNA in CLIP assays (*Figure 2A*). For each protein, the specificity of Ψ binding was represented quantitatively by plotting the number of Gag-bound gRNA-derived reads derived from the Ψ region in CLIP experiments as a fraction of all gRNA-derived reads (*Figure 2—figure supplement 1*). First, we performed CLIP experiments with CANC mutants with deletions introduced into the NC domain, whereby the N-terminal zinc finger (CANC dZF1), the C-terminal zinc finger (CANC dZF2), or both zinc fingers (CANC dZF) were deleted (*Figure 2A*). The deletion of either zinc finger in the NC domain caused a marked reduction in specific Ψ binding, while the deletion of both zinc fingers nearly completely abolished specific Ψ binding, despite the presence of similar levels of protein for each of the mutant CANC proteins (*Figure 2B*, *Figure 2—figure supplements 1–2*). This observation is consistent with earlier findings that NC is required for the specific interaction between Gag and Ψ (*Aldovini and Young, 1990*; *Didierlaurent et al., 2011*; *Dorfman et al., 1993*; *Gorelick et al., 1990*).

Next, to test whether an isolated NC domain was sufficient for specific binding to Ψ, we deleted the region encoding the CA domain from the CANC construct, thereby generating a construct that encodes NC (along with the SP2-p6 C-terminal extension) with an N-terminal 3xHA tag. CLIP experiments using this construct revealed that NC was well expressed but was not sufficient to bind specifically to Ψ. Instead, low read counts, distributed across the entire HIV-1 genome, were obtained (*Figure 2B*, *Figure 2—figure supplements 1 and 2*). Indeed, the deficit in specific Ψ binding associated with CA deletion was as profound as the deficit associated with deletion of both NC zinc fingers (*Figure 2B*).

Since CANC may form multimers, we considered the possibility that multimerized NC may be sufficient to enable Ψ-specific RNA binding. Thus, to test whether CA-induced multimerization could be recapitulated by fusion to heterologous protein-protein interaction domains that can drive hexamerization, we substituted the CA domain in the CANC constructs. The C terminal residues of CA and SP1 important for CASP1 6-helix bundle formation (*Accola et al., 1998*; *Datta et al., 2011*; *Schur et al., 2016*; *Wagner et al., 2016*) were kept intact, and we placed GCN4pII (GCN4pII-SP1-NC) or GCN4pAA (GCN4pAA-SP1-NC) leucine zippers (*Harbury et al., 1994*; *Liu et al., 2006*), or ccHex2 synthetic peptide (ccHex2-SP1-NC) between the 3xHA tag and the CASP1 helix (*Figure 2A*). The ccHex2 synthetic peptide forms hexameric parallel a-helical barrel coil-coils (*Thomson et al., 2014*). To test for multimerization in cells, we treated HEK293T cells transfected with plasmids expressing CANC and derivatives thereof with 1,6-Bismaleimidohexane (BMH), a cell-permeable maleimide crosslinker that mediates irreversible conjugation between sulfhydryl groups (*Dewson, 2015*). Protein multimerization was then assessed by western blot analysis of cell lysates with an anti-NC antibody. The chimeric proteins were expressed at level similar to CANC and were able to form multimers, up to and including hexamers, as indicated by the appearance of BMH crosslinked species (*Figure 2C*, *Figure 2—figure supplements 2 and 3*). Nevertheless, the chimeric, multimeric NC fusion proteins did not bind specifically to Ψ (*Figure 2D*, *Figure 2—figure supplement 1*); instead, they exhibited low-level indiscriminate binding across the gRNA, similar to the isolated monomeric NC domain (*Figure 2B*, *Figure 2—figure supplement 1*). These results suggest that CA is required for the specific interaction between CANC and Ψ and that CA provides some function beyond simple multimerization.

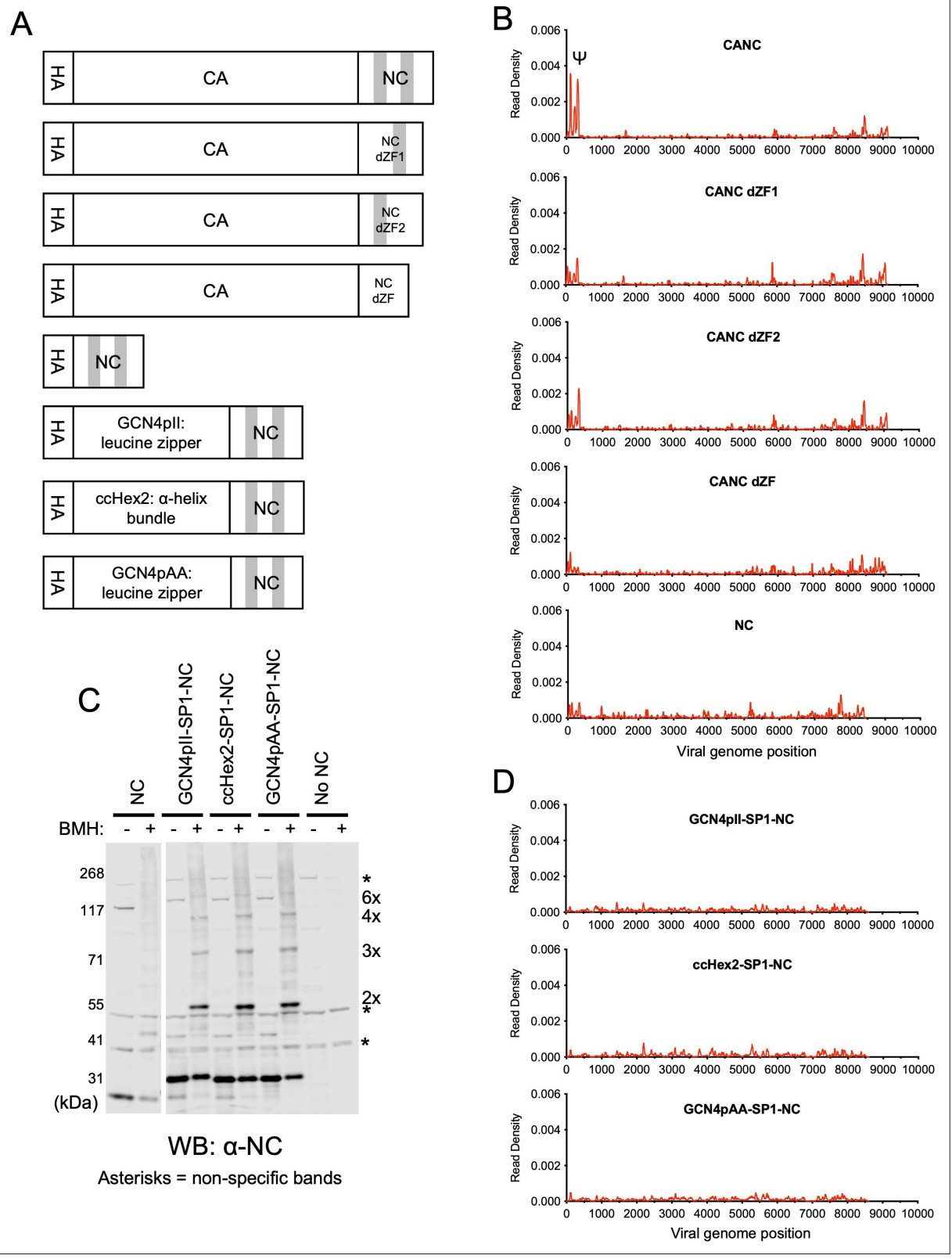

**Figure 2.** RNA binding specificity of nucleocapsid (NC) zinc-finger deletion mutants, monomeric NC, and artificially multimerized NC proteins. (**A**) Schematic representation of the constructs used. Shaded regions indicate NC zinc fingers. (**B**) Read density distribution on viral RNA from crosslinking immunoprecipitation (CLIP) experiments in which constructs encoding CANC, and mutant derivatives with deletions of the zinc finger 1 (CANC dZF1), zinc finger 2 (CANC dZF2), or both zinc fingers (CANC dZF) were used. Alternatively, a construct (NC) in which capsid (CA) was deleted

*Figure 2 continued on next page*

*Figure 2 continued*

was used. Each chart represents at least two independent experiments, and the average read density is plotted. (**C**) Western blot analysis of Gag-derived proteins following chemical crosslinking in living cells using 1,6-Bismaleimidohexane (BMH) prior to cell lysis. Proteins were detected with an anti-NC antibody. (**D**) Read density distribution on viral RNA from CLIP experiments in which constructs encoding chimeric NC proteins with artificial multimerizing domains were used. Each chart represents at least two independent experiments, and the average read density is plotted.

The online version of this article includes the following source data and figure supplement(s) for figure 2:

**Source data 1.** Uncropped and labeled blots for *Figure 2C*.

**Source data 2.** Read density data for *Figure 2B* and *Figure 2D*.

**Figure supplement 1.** Quantification of Ψ binding specificity by CANC, nucleocapsid (NC), and artificially multimerized NC proteins.

**Figure supplement 2.** Quantification of protein expression levels of crosslinking immunoprecipitation (CLIP) constructs.

**Figure supplement 2—source data 1.** Uncropped and labeled blots *Figure 2—figure supplement 2*.

**Figure supplement 3.** Quantitative analysis of BMH crosslinked species in *Figure 2C*.

## CA driven assembly of a nascent immature Gag lattice is required for specific Ψ recognition

To investigate how CA drives the specific interaction between CANC and Ψ, we designed 21 CA substitution mutants. These substitution mutants were selected based on the structures of HIV-1 immature Gag lattice and mutagenesis studies that revealed residues important for mature or immature CA assembly (*Forshey et al., 2002*; *Ganser-Pornillos et al., 2004*; *Schur et al., 2016*; *von Schwedler et al., 2003*; *Wagner et al., 2016*). 10 substitution mutants targeted the CA N-terminal domain (NTD): R18A/N21A, A22D, E28A/E29A, P38A, A42D, E45A, D51A, R100A/S102A, T107A/T108A, and T110A/Q112A, and 11 substitution mutants targeted the CA C-terminal domain (CTD) and SP1 region: K158A, W184A/M185A, D197A, Q219A, G222A, P224A, K227A, R229A, SP1 M4A, SP1 T8I, and SP1 T12A (*Figure 3A*).

These substitutions targeted interfaces at either the twofold, threefold, or sixfold axes of symmetry, where interactions required for the formation of the complete immature Gag lattice occur (*Figure 3B–E*). Each mutant was subjected to three types of experiment: first, in vivo chemical crosslinking was performed to test propensity of mutant CANC proteins to multimerize. Second, virus particle production experiments were done in which the CA substitutions were introduced to full-length HIV-1 NL4-3 and Gag processing/extracellular virion formation tested. Finally, the panel of CANC mutants was subjected to CLIP experiments to test for specific binding to Ψ.

In the in vivo CANC chemical crosslinking experiments, we observed a ladder-like pattern of bands indicating the cytoplasmic formation of higher-order multimers for the unmanipulated CANC, as well as certain CANC mutants (*Figure 4A*, *Figure 4—figure supplement 1*). Indeed, the SDS PAGE/western blot approach permitted the detection of crosslinked species containing up to ~8–10 CANC subunits. Based on these results, the CA mutants could be divided into three categories: (i) mutants with no deficits in the formation of higher-order multimers, specifically R18A/N21A, P38A, E45A, Q219A, SP1 T8I, and SP1 T12A; (ii) mutants with moderate deficits in multimerization, characterized by reduced abundance of higher order crosslinked species; these mutants included E28A/E29A, D51A, R100A/S102A, T107A/T108A, T110A/Q112A, and R229A; and (iii) mutants with severe defects in multimerization, characterized by the appearance of CANC monomers as the major species even after crosslinking; these mutants included A22D, A42D, W184A/M185A, D197A, G222A, P224A, and SP1 M4A (*Figure 4A*, *Figure 4—figure supplement 1*). The phenotype of some mutants in the in vivo chemical crosslinking experiment was expected based on the HIV Gag/CANC immature lattice structures and previous in vitro assembly studies (*Schur et al., 2016*; *von Schwedler et al., 2003*; *Wagner et al., 2016*). The effect of the mutations varied, and substitution of residues with more bulky residues (e.g. A22D and A42D) in the CA NTD was sometimes more disruptive to CA multimerization than the substitution of residues with less bulky residues (e.g. R18A/N21A, E28A/E29A, and T110A/Q112A). Nevertheless, substitutions that selectively targeted the twofold interface (W184A/M185A), the threefold interface (A22D and A42D), or the sixfold interface (R100A/S102A, T110A/Q112A, T107A/T108A, D197A, G222A, P224A, and SP1 M4A) each caused defects in CANC multimerization, consistent with previous studies showing that these interfaces are required for the immature lattice assembly (*Accola*

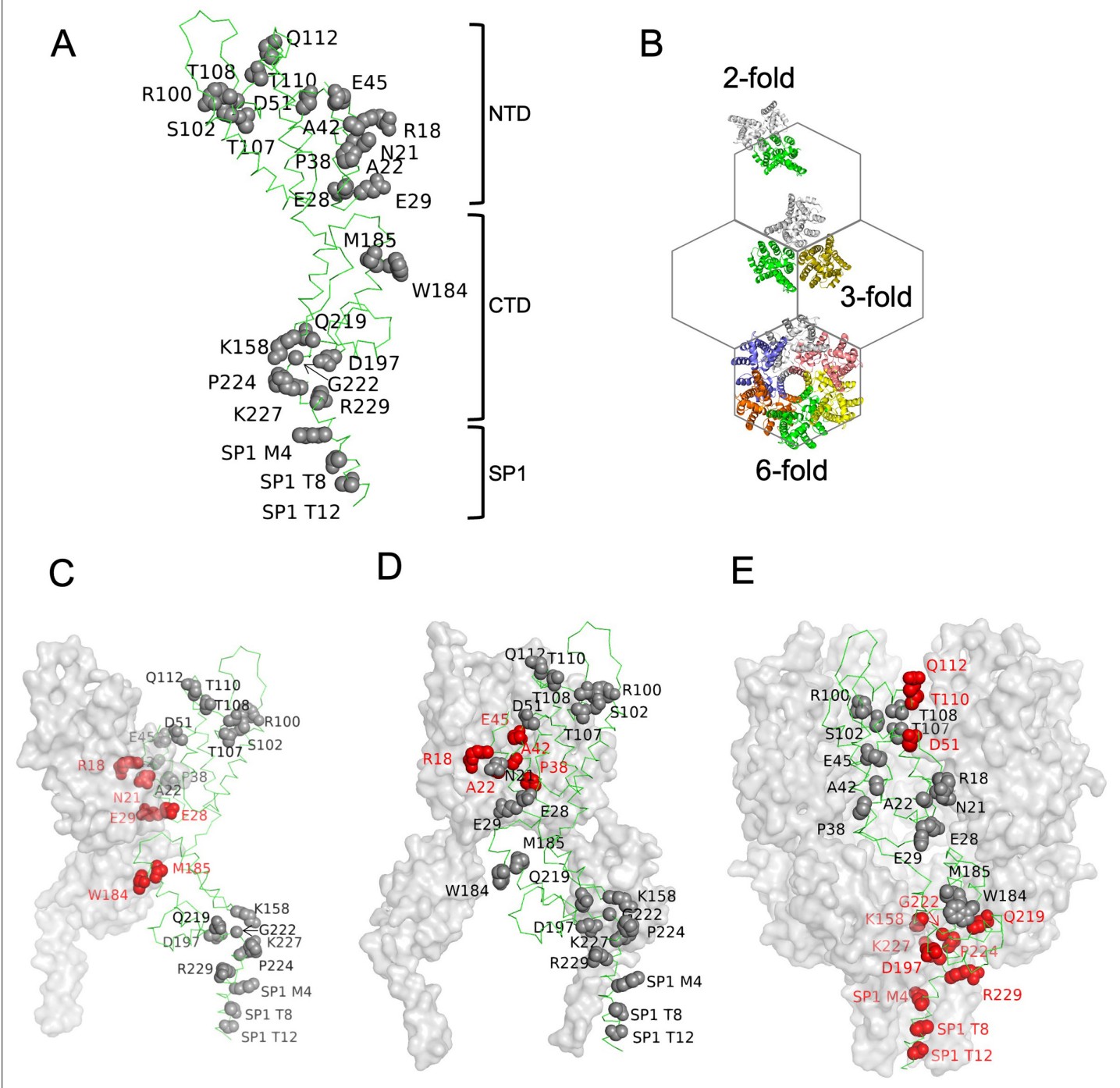

**Figure 3.** Amino acids substitutions targeting capsid (CA) interfaces in the immature Gag lattice. (**A**) Amino acids selected for substitution are depicted onto a CA monomer component of the immature HIV-1 Gag lattice (adapted from PDB 7ASH). Residues subjected to substitution are shown in ball and stick presentation. (**B**) A schematic of the immature hexagonal CA lattice showing twofold, threefold, and sixfold symmetry contacts. (**C–E**) Depiction of substitutions targeting the twofold (**C**), threefold (**D**) , and sixfold (**E**) CA-CA interaction interfaces. One monomer of CASP1 is labeled, and the residues at the respective CA-CA interaction interfaces are colored in red. Other monomers are shown in gray surface presentation.

et al., 1998; Datta et al., 2011; Dick et al., 2018; Liang et al., 2002; Mallery et al., 2021; Schur et al., 2016; von Schwedler et al., 2003; Wagner et al., 2016).

To test whether the CA mutants could support lattice assembly that would lead to the generation of extracellular virions, we introduced the same set of CA substitutions into a full-length infectious HIV-1 NL4-3 proviral construct and assessed virion production by proviral plasmid transfected HEK293T

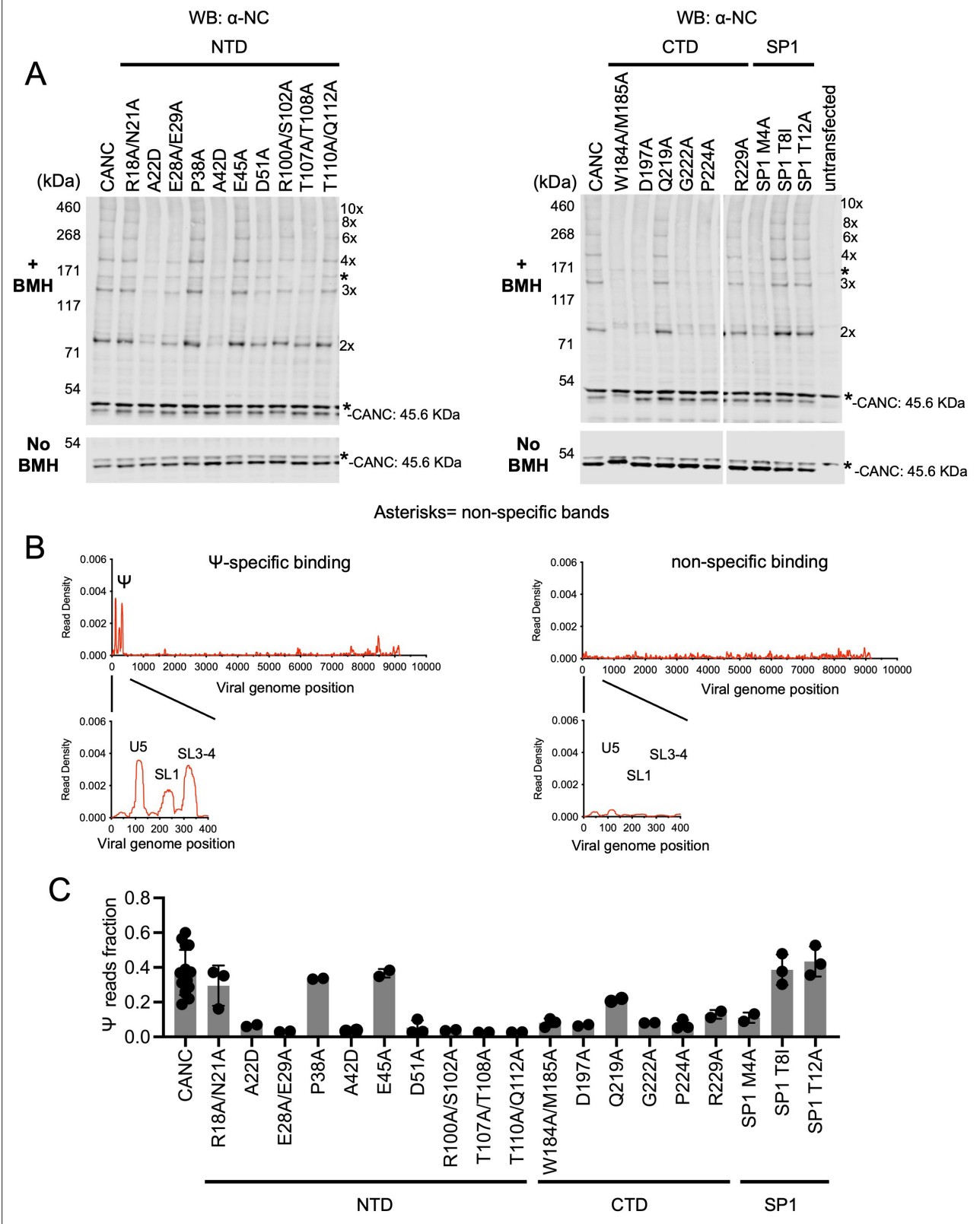

**Figure 4.** Effects of caspid (CA) substitutions on the in vivo multimerization and RNA binding properties of CANC proteins assessed by crosslinking experiments. (**A**) Western blot analysis of CANC proteins following chemical crosslinking in living cells using BMH prior to cell lysis. Proteins were detected with anti-nucleocaspid (NC) antibody. (**B**) Typical results of crosslinking immunoprecipitation (CLIP) experiments of using CANC mutants. The left panel indicates the specific Ψ binding by CANC and certain mutants thereof. Mutants in this category include: R18A/N21A, P38A, E45A,

*Figure 4 continued on next page*

*Figure 4 continued*

Q219A, SP1 T8I, and SP1 T12A. The right panel indicates lack of specific Ψ binding; mutants in this category include: A22D, E28A/E29A, A42D, D51A, R100A/S102A, T107A/T108A, T110A/Q112A, W184A/M185A, D197A, G222A, P224A, R229A, and SP1 M4A. CLIP results for each mutant are shown in *Figure 4—figure supplement 3*. (**C**) Quantification of Ψ binding specificity of CANC mutants. The decimal fraction of reads, calculated by dividing the number of reads that mapped in the Ψ region of the genome (coordinates: 101–356) by the total number of reads that mapped to the viral genome, is plotted. Each dot represents data from an independent experiment. Error bars indicate the SD of all independent experiments.

The online version of this article includes the following source data and figure supplement(s) for figure 4:

**Source data 1.** Uncropped and labeled blots *Figure 4A*.

**Figure supplement 1.** Quantitative analysis of BMH crosslinked species in *Figure 4A*.

**Figure supplement 2.** Western blot analysis of HIV-1 NL4-3 caspid (CA) mutants in particle production experiments.

**Figure supplement 2—source data 1.** Uncropped and labeled blots from *Figure 4—figure supplement 2*.

**Figure supplement 3.** Effect of caspid (CA) mutations on CANC RNA binding specificity.

**Figure supplement 3—source data 1.** Read density data for *Figure 4—figure supplement 3*.

**Figure supplement 4.** Reverse transcription-quantitative polymerase chain reaction (RT-qPCR) quantification of copies of unspliced gRNA in virions generated by HIV-1 NL4-3 and caspid (CA) mutants in *Figure 4—figure supplement 2*.

**Figure supplement 4—source data 1.** Uncropped and labeled blots from *Figure 4—figure supplement 4A*.

---

cells. This analysis revealed that mutants such as R18A/N21A, P38A, E45A, Q219A, SP1 T8I, and SP1 T12A generated virions at, or close to, wild-type levels, while other mutants either generated reduced levels of virions or failed to generate virions (*Figure 4—figure supplement 2*). These results are consistent with previous studies (*von Schwedler et al., 1998*; *von Schwedler et al., 2003*). Results from the in vivo CANC chemical crosslinking experiments and virion production experiments showed a correlation between the ability of a given CA mutant to affect CANC multimerization and virion production: mutants that formed high-order CANC multimers produced more virions than mutants in which CANC multimerization was impaired (*Figure 4A*, *Figure 4—figure supplements 1–2*).

When CANC proteins encoding the same panel of mutants were subjected to CLIP experiments, the mutants could be divided into two categories. One group of mutants bound to Ψ with high specificity; this group included R18A/N21A, P38A, E45A, Q219A, SP1 T8I, and SP1 T12A. A second group included all mutants other than the aforementioned group of six; this group of CANC mutants fail to bind to Ψ specifically (*Figure 4B and C* and *Figure 4—figure supplement 3*). Comparison of the behaviors of the CANC mutants in CLIP experiments and in vivo cytoplasmic chemical cross-linking experiments revealed a correlation between CA multimerization and Ψ-specific binding: CANC mutants that were readily able to form high-order multimers in the cytoplasm bound to Ψ with high specificity, similar to the wild-type CANC (in the cases of R18A/N21A, P38A, E45A, Q219A, SP1 T8I, and SP1 T12A). Conversely, mutants that had deficits in cytoplasmic multimerization failed to bind specifically to Ψ. Notably, impairment of Ψ binding imposed by CA substitutions occurred irrespective of which interface (twofold, threefold, or sixfold) was targeted. Overall, these experiments suggest that the ability of CANC to form a nascent immature lattice is important for specific recognition of Ψ.

To examine the relationship between cytosolic CANC binding to Ψ and packaging of vRNA into extracellular virions, we measured the vRNA:Gag ratio for all the CA mutants that generated some level of extracellular particles. This included some mutants that, in the context of CANC, exhibited impaired cytoplasmic multimerization and specific Ψ binding in the cytoplasm (A22D, E28A/E29A, and D51A) or for which Ψ binding appeared marginally impaired (Q219A and R229A). Nevertheless, these mutants were able to generate extracellular particles in the context of full-length HIV-1 NL4-3. There were only minor variations in the vRNA:Gag ratios for these mutants, suggesting that they are able to package vRNA (*Figure 4—figure supplement 4*). However, given that these mutants generated extracellular virions, they must, by definition, have been able to assemble. An essential step in assembly is the generation of an immature Gag lattice. Thus, these mutant full-length Gag proteins must have assembled an immature Gag lattice, despite the fact that the CANC proteins do not appear to assemble into higher-order multimers in the cytoplasm. We posit that the CA mutants that generate virions in the context of full-length virus but do not generate high-order CANC multimers in the cytosol, harbor partial defects in immature lattice formation that are evident in the context of CANC in the cytosol but are at least partly suppressed when Gag is targeted to membrane in the context

of a full-length Gag protein. This idea is consistent with a notion proposed by *O'Carroll et al., 2012*, who invoke functional redundancy between membrane binding, CA-CA interaction, and RNA binding in driving HIV-1 particle assembly. Notably, these results do not alter the interpretation that immature lattice assembly is required for Ψ binding but do suggest that immature lattice assembly can occur either in the cytosol or at the plasma membrane to enable Ψ recognition.

### Deficits in Ψ recognition are exhibited by inositol hexakisphosphate binding-deficient mutants but are restored in second-site revertants

The abundant intracellular small molecule inositol hexakisphosphate (IP6) coordinates two rings of positively charged lysine residues formed by K158 and K227 at the base of the CA hexamer (*Dick et al., 2018*). As such, it is required for the assembly of the immature Gag lattice in cells. To corroborate the above findings that the formation of a nascent immature lattice is required for specific Ψ binding, we performed CLIP experiments using two mutants (CANC K158A and CANC K227A) that are impaired for IP6 coordination and are thus incapable of particle assembly. Both CANC K158A and CANC K227A mutants exhibited deficits in Ψ binding (*Figure 5A*). A second-site revertant of K158A and K227A, specifically a lattice stabilizing substitution in SP1(SP1 T8I), restores virion assembly and viral infectivity (*Mallery et al., 2021*; *Poston et al., 2021*). Notably, introduction of this second-site substitution (in CANC K158A/SP1 T8I and CANC K227A/SP1 T8I) enabled otherwise Ψ-binding defective CANC proteins to bind Ψ with high specificity (*Figure 5B and C*, and *Figure 5—figure supplement 1*). Thus, these results reinforce the conclusion that the specific binding to Ψ requires the correct assembly of the nascent viral immature CA lattice.

## Discussion

We conclude that the initiation of the assembly of the immature HIV-1 Gag lattice in infected cells is required for the maintenance of the interaction between Gag/CANC and the Ψ element of the gRNA. Nevertheless, questions remain about the precise number of Gag/CANC monomers that are required to assemble to enable specific Ψ binding. Imaging studies of the HIV-1 virion assembly process suggest that a small number of Gag molecules (below the limit of detection by fluorescent microscopy) is involved in initial Gag/gRNA complex formation (*Hendrix et al., 2015*; *Jouvenet et al., 2009*; *Kutluay and Bieniasz, 2010*). However, the ability of CANC to multimerize into higher-order multimers (such as 10-mers or greater) in our in vivo chemical crosslinking experiments predicted specific Ψ binding, and the twofold, threefold, and sixfold CA interaction interfaces are all required. These findings suggest that perhaps high-order multimers of CANC may be needed to recognize Ψ. The artificially multimerized cytoplasmic NC domains were able to generate hexamers, but unable to bind Ψ, consistent with the notion that higher order multimerization is required for Ψ binding. Alternatively, it is also possible that CA is required to precisely position NC domains within a hexamer to permit specific Ψ binding.

Several lines of evidence from previously published work are consistent with the proposition that high-order Gag multimers bind to Ψ: (1) SHAPE and XL-SHAPE experiments identified at least 10 potential Gag/NC interaction sites in gRNA which are important for packaging (*Kenyon et al., 2015*; *Wilkinson et al., 2008*); (2) electrophoretic mobility shift experiments suggest that at least 20 Gag molecules were needed to form a complete Gag/leader gRNA dimer complex (*D'Souza et al., 2021*); (3) studies of NC and Ψ interactions with nuclear magnetic resonance (NMR) and isothermal titration calorimetry (ITC) suggest that there are at least two dozen high-affinity NC-binding sites in a dimerization competent form gRNA (*Ding et al., 2020*). Further studies are thus needed to determine the precise stoichiometry between Gag and Ψ interaction at the initiation of Ψ binding and particle assembly.

There were some unexpected findings from our chemical crosslinking experiments. For example, the W184A/M185A mutant was mainly monomeric under the crosslinking conditions in our assay, despite the fact that W184/M185 are located at an inter-hexamer interface rather than an intra-hexamer interface. In a previous report, a CA quadruple mutant A14C/E45C/W184A/M185A was able to form a hexamer under reducing conditions (*Pornillos et al., 2009*). Possible explanations for this discrepancy include: (i) the A14C/E45C/W184A/M185A CA mutant was purified and crosslinked in vitro, while in our study, W184A/M185A crosslinking was attempted in cells; (ii) the A14C/

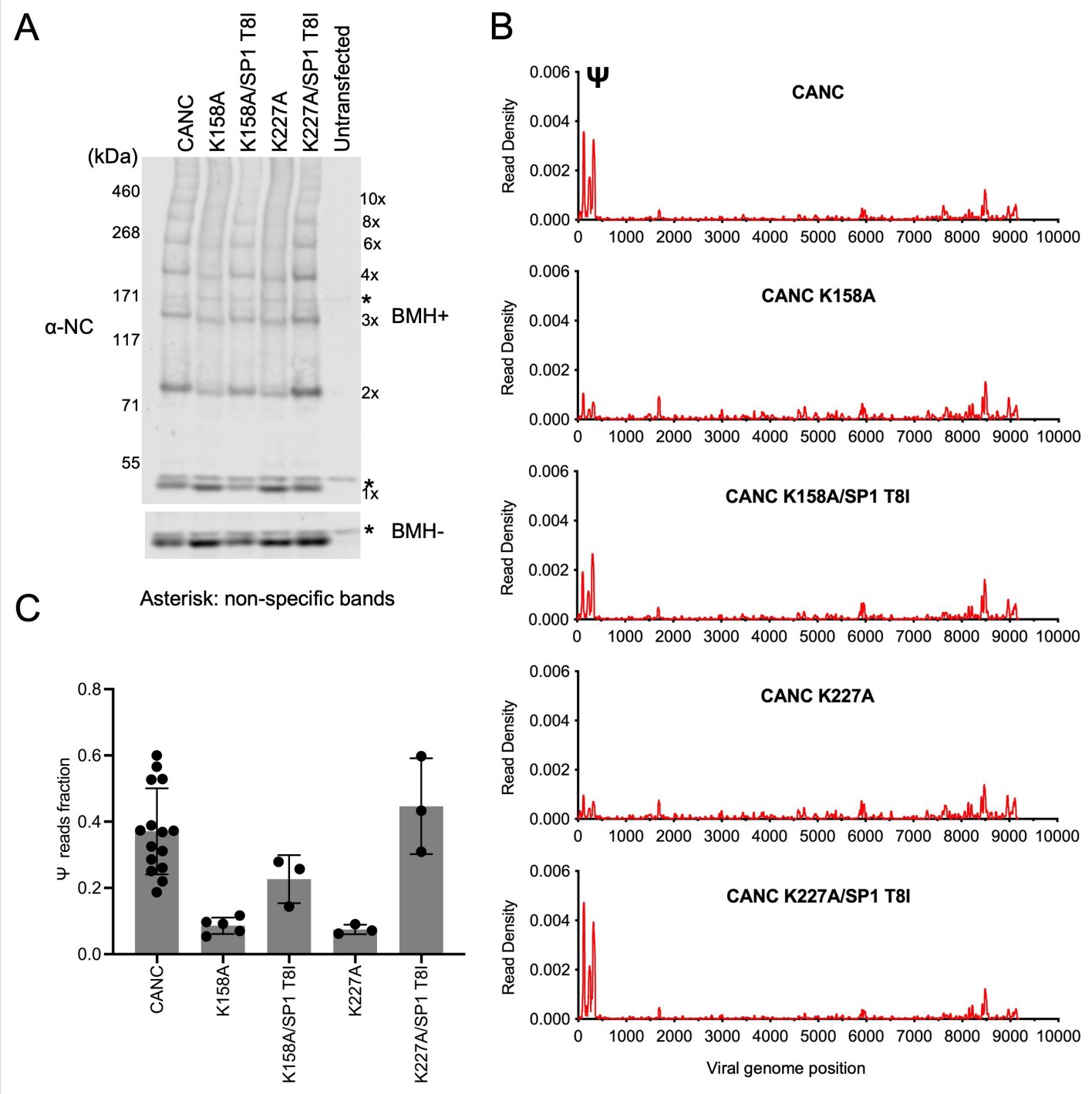

**Figure 5.** Analysis of multimerization and RNA binding by hexakisphosphate (IP6)-binding deficient mutants and second-site revertants. (**A**) Western blot analysis of mutants CANC K158A, CANC K227A, and the corresponding second-site revertants CANC K158A/SP1 T8I and CANC K227A/SP1 T8I following chemical crosslinking in living cells using BMH prior to cell lysis. Proteins were detected with anti-NC antibody. (**B**) Read density distribution on viral RNA from crosslinking immunoprecipitation (CLIP) experiments in which constructs encoding CANC, CANC K158A, CANC K158A/SP1 T8I, CANC K227A, and CANC K227A/SP1 T8I were used. Each chart represents at least two independent experiments, and the average read density is plotted. (**C**) Quantification of Ψ binding specificity of the mutants in panel (**B**). The decimal fraction of reads, calculated by dividing the number of reads that mapped in the Ψ region of the genome (coordinates: 101–356) by the total number of reads mapped to the viral genome, is plotted.

The online version of this article includes the following source data and figure supplement(s) for figure 5:

**Source data 1.** Uncropped and labeled blots from *Figure 5A*.

*Figure 5 continued on next page*

*Figure 5 continued*

**Source data 2.** Read density data for *Figure 5B, C*.

**Figure supplement 1.** Quantitative analysis of BMH crosslinked species in *Figure 5A*.

E45C/W184A/M185A CA mutant assembled through mature lattice interactions, whereas herein the W184A/M185A mutant was constrained by N- and C-terminal extensions to form the immature lattice, which involves different interfaces. Another unexpected finding from our in vivo chemical crosslinking studies was that certain mutants, such as A22D and A42D, were mainly monomeric under in vivo crosslinking conditions even though A22D and A42D might be expected to form dimers or hexamers because these substitutions disrupt the three-fold CA-CA interaction interface rather than the twofold or the sixfold CA-CA interaction interfaces (*Schur et al., 2016*; *Wagner et al., 2016*). This result, along with the fact that the dimer interface mutants E28A/E29A and W184A/M185A mutants were also mainly monomeric in cells, suggests that both dimer and trimer CA interfaces contribute to the formation of early lattice assembly intermediates. Moreover, mutants that failed to form crosslinkable dimers via inter-hexamer contacts also failed to form hexamers, suggesting that inter-hexamer contacts are important for hexamer assembly. Thus, previous models (*Grime and Voth, 2012*; *Tomasini et al., 2018*; *Tsiang et al., 2012*) which proposed that trimer-of-dimers of CA are basic building block of the HIV-1 immature lattice are not consistent with our crosslinking results. Overall, the results of the crosslinking experiments suggest that all the three CA-CA interaction interfaces (twofold, three-fold, and sixfold interaction interfaces) contribute simultaneously to immature lattice formation since disruptions of either twofold (W184A/M185A), threefold (A22D and A42D), or sixfold (D197A, G222A, P224A, and SP1 M4A) CA-CA interaction interfaces lead to overall defects rather than the formation of discrete low-order multimeric species.

Recently, Duchon et al. used complementation approaches with Gag proteins whose multimerization was driven by leucine-zippers to show that membrane anchoring can increase the efficiency of RNA packaging, whether driven by interactions between NC and Ψ or by an artificial RNA binding protein:RNA target pair (*Duchon et al., 2021*). Nevertheless, our study shows that membrane binding is not required for CANC-Ψ interaction in the cytoplasm of cells. Given that the localization of Gag can change with its concentration in cells, it is possible that the actual site of Gag-Ψ interaction could also change as Gag accumulates in infected cells. Notably, we found that certain CA mutations that conferred multimerization and Ψ binding deficits that were evident in the context of cytosolic CANC could generate some level of extracellular particles. In these cases, the vRNA:Gag ratio in extracellular virions was close to that of WT. This finding suggests that membrane binding can suppress both the Gag multimerization and Ψ binding deficits exhibited by some CA mutants and, thus, that Ψ binding can, in principle, occur at the plasma membrane or in the cytosol, provided that higher order multimerization occurs.

Overall, our study shows that CA is essential for the specific interaction between HIV-1 Gag and Ψ. CA is therefore key, not only for controlling the morphology of HIV-1 particle assembly but also for selective viral genome packaging. Based on the evidence that HIV-1, HIV-2, and SIV Gag can co-assemble and package each other's genome (*Al Shamsi et al., 2011*; *Franke et al., 1994*; *Motomura et al., 2008*; *Rizvi and Panganiban, 1993*) and the conservation of primary and tertiary Gag structures between these viruses, the insights gained from this study are likely applicable across primate lentiviruses. Whether the findings described herein constitute a more generalized principle, and that formation of nascent CA or NC lattices enables specific interactions between virion proteins and genomes in the assembly of other viruses with RNA genomes, remains to be determined.

## Materials and methods

**Key resources table**

| Reagent type (species) or resource | Designation | Source or reference | Identifiers | Additional information |
|---|---|---|---|---|
| Cell line (*Homo sapiens*) | 293T | ATCC | CRL-3216 | Periodically checked for mycoplasma and retrovirus contamination not authenticated since purchased directly from ATCC |

*Continued on next page*

*Continued*

| Reagent type (species) or resource | Designation | Source or reference | Identifiers | Additional information |
|---|---|---|---|---|
| Transfected construct (HIV-1) | pNL4-3 and CA mutants introduced in pNL4-3 | This paper | | For viral production assay and RT-qPCR gRNA quantification assay |
| Transfected construct (HIV-1) | NL4-3 (MA-3xHA/PR-) and its derivatives | This paper | | For CLIP experiments and BMH crosslink experiments |
| Antibody | Mouse monoclonal anti-HIV-1 p24CA | NIH AIDS Reagent Program | 183-H12-5C | WB (1:100) |
| Antibody | Rabbit polyclonal anti-HIV-1 Nucleocapsid | This paper | | WB (1:2000) |
| Commercial assay or kit | Power SYBR Green RNA-to-CT 1-Step Kit | ThermoFisher | Cat# 4389986 | |
| Chemical compound and drug | BMH (bismaleimidohexane) | ThermoFisher | Cat# 22330 | |
| Software and algorithm | Prism | Graphpad | | For CLIP data analysis graphing |
| Software and algorithm | Image Studio | LI-COR Biosciences | | For western blot band quantification |
| Software and algorithm | Cutadapt | https://doi.org/10.14806/ej.17.1.200 | | For CLIP data processing |
| Software and algorithm | FASTX toolkit | http://hannonlab.cshl.edu/fastx_toolkit | | For CLIP data processing |
| Software and algorithm | Bowtie | https://doi.org/10.1186/gb-2009-10-3-r25 | | For CLIP data processing |
| Software and algorithm | SAMTools | https://doi.org/10.1093/bioinformatics/btp352 | | For CLIP data processing |

## Plasmids and cells

Constructs for CLIP experiments were generated based on a previously described HIV-1 NL4-3 (MA-3xHA/PR-) proviral construct, which contains a 3xHA tag within the stalk region of MA (between residues 127 and 128) and an inactivating mutation (D81A) in the viral protease (*Kutluay et al., 2014*). The CANC constructs were generated by changing the start codon (ATG) of the MA domain of NL4-3 (MA-3xHA/PR-) to ACG to abrogate the expression of MA, thus translation begins at an AUG codon at the amino terminus of the 3xHA tag. CANC variants were generated by overlapping PCR to introduce substitutions in the CA domain. The NC construct was generated by overlapping PCR to delete the CA domain in the CANC construct. GCN4pII, GCN4pAA, and ccHex2 chimeric constructs were generated by overlapping PCR to replace the CA domain in the CANC construct (but retaining the CA C-terminal residues '$_{223}$GPGHKARVL$_{231}$' intact for CASP1 helix formation) with GCN4pII or GCN4pAA leucine zippers or with ccHex2 synthetic peptide. For GCN4pII-SP1-NC constructs, the amino-acid sequence and junctions are as follows: **RMKQIEDKIEEILSKIYHIENEIARIKKLIGER**TS*GPGHKARVL*, where the GCN4pII is in bold, the CA C-terminal residues in italic, and a two amino acids 'TS' linker in between. For GCN4pAA-SP1-NC constructs, the amino-acid sequence and junctions are as follows: **MKVKQLADAVEELASANYHLANAVARLAKAVGER**GS*GPGHKARVL*, where the GCN4pAA is in bold, the CA C-terminal residues in italic, and a two amino acids 'GS' linker in between. For ccHex2-SP1-NC constructs, the amino-acid sequence and junctions are as follows: **GEIAKSLKEIAKSLKEIAWSLKEIAKSLKG**S*GPGHKARVL*, where the ccHex2 is in bold, the CA C-terminal residues in italic, and a single amino acid 'S' in between. In the virion production assay, substitution mutations were introduced into the CA domain of the HIV-1 NL4-3 wild-type proviral plasmid. HEK293T cells were transfected with these constructs for CLIP experiments, in-cell chemical crosslinking experiments, and virion production experiments.

## Crosslinking immunoprecipitation coupled with sequencing

HEK293T cells at 90% confluency in a 15 cm dish were transfected with 25 µg proviral plasmids using polyethylenimine (PEI). Cell culture media was replaced with fresh media 10–14 hr after transfection. Ribonucleoside analog 4-thiouridine (4SU) was added to the media 12–14 hr before UV crosslinking

at a final concentration of 100 µM. On the day of crosslinking, cells were rinsed with PBS once and crosslinked with 500 mJ/cm$^2$ UV ($\lambda$ =365 nm) in a Boekel UV crosslinker. After UV crosslinking, cells were resuspended in 15 ml PBS, and cell pellets were collected at 500 × $g$ for 5 min of centrifugation. Cell pellets were stored at –80°C until use. The CLIP procedure was modified based on previously published protocols (*Kutluay et al., 2014*; *Shema Mugisha et al., 2020*). Modifications are in two areas: (i) a 3′ adapter containing an infrared dye (*Zarnegar et al., 2016*) was used instead of a 3′ adapter labeled with $^{32}$P; (ii) the 3′ adapter was ligated to protein/RNA complexes on antibody-conjugated Dynabeads (ThermoFisher 10004D) instead of ligating to purified RNAs. Rnase A/T1 mix (ThermoFisher EN0551) was used at a dilution of 1:100 to digest RNAs in the cell lysate before immunoprecipitation. For immunoprecipitation of CLIP constructs, we used anti-HA mouse IgG (BioLegend, 901503). Alternatively, in experiments with artificially multimerized NC proteins, we used a custom rabbit polyclonal IgG antibody raised against the NC domain of the HIV-1 NL4-3.

## Bioinformatic analysis

The CLIP library was sequenced with Illumina Nextseq 500 platform. Raw fastq reads were processed with Cutadapt (*Martin, 2011*), and reads that were fewer than 20 nt, did not contain the 3′ adapter, or contained ambiguous nucleotides were excluded. Barcodes were collapsed and trimmed with the FASTX toolkit (http://hannonlab.cshl.edu/fastx_toolkit) prior to mapping. Reads were mapped using Bowtie (*Langmead et al., 2009*) to the corresponding viral genomes in CLIP experiments. SAMtools (*Li et al., 2009*) and in-house scripts (*Kutluay et al., 2014*) were used to generate counts of each base in the viral genomes. Read density was calculated from the counts of each base divided by the total counts of bases mapped to the corresponding viral genome. Read density graphs were generated with GraphPad Prism 9 (https://www.graphpad.com/). The $\Psi$ binding specificity was quantified by dividing the number of reads mapped in the $\Psi$ region by the total number of reads that mapped to the viral genome.

## In vivo BMH crosslinking

The use of cysteine-specific crosslinker BMH to study CANC interactions in cells was based on earlier studies of CANC assembly in vitro (*Hansen and Barklis, 1995*; *McDermott et al., 1996*). In our study, the BMH crosslinking method was modified to study CANC interactions in cells. Specifically, 1 × 10$^6$ HEK293T cells were seeded in each well of a six-well plate 1 day prior to transfection. Then 2 µg of pCANC and variants thereof were transfected with PEI, and culture media was replaced with fresh media 7–10 hr after transfection. Cells were washed once with PBS 20–24 hr post-transfection and suspended with 1 ml PBS/EDTA (5 mM; cell density was around 2.5–3 × 10$^6$/ml). Next, 200 µl cell suspension was used for BMH crosslinking whereby 5 µl of freshly dissolved BMH (40 mM stock in Dimethyl sulfoxide (DMSO), ThermoFisher 22330) was added to 200 µl cell suspension (final BMH concentration 1 mM) to start the crosslinking reaction. DMSO mock-treated reactions were simultaneously performed. The reactions were incubated in the dark at room temperature for 1 hr. Then, 7 µl 1 M Dithiothreitol (DTT) was added to the reactions for 15 min of incubation to stop the reactions. Next, 72 µl 4xNuPAGE LDS Sample Buffer (ThermoFisher NP0008) was added to the reaction followed by 15 min incubation. The samples were sonicated and heated at 72°C for 10 min before loading 20 µl/well on NuPAGE 3–8% Tris-Acetate gels (ThermoFisher, EA03785BOX) alongside high-molecular weight markers (ThermoFisher LC5699). The gel was run at 150 Volts for 60 min. Gels were transferred in the transfer buffer (ThermoFisher NP0006) with 20% ethanol at 35 Volts for 90 min to nitrocellulose membranes (Cytiva 10600002). After transfer, membranes were probed using a BlotCycler (Precision Biosystem) with mouse monoclonal anti-HIV-1 p24CA (183-H12-5C, NIH AIDS Reagent Program) primary antibody and custom rabbit polyclonal anti-HIV-1 NC primary antibody. IRDye 680RD Donkey anti-Mouse IgG Secondary Antibody (Licor P/N: 926–68072) and IRDye 800CW Donkey anti-Rabbit IgG Secondary Antibody (Licor P/N: 926–32213) were used for detection. Membranes were imaged at the Licor Odyssey imaging system.

## Virion production assay

HEK293T cells (3 × 10$^5$) were seeded in each well of a 24-well plate. Proviral plasmids (1 µg HIV-1 NL4-3 wild type or CA mutant) were transfected with PEI the next day. Culture media was replaced with fresh media 1 day after transfection. At 40–48 hr after transfection, 600 µl virion supernatant from

each sample was filtered with a 0.2 µm filter and gently placed above an equal volume of 20% sucrose/PBS cushion in a 1.5 ml Eppendorf tube. Virions were pelleted by centrifugation at 14,000 rpm for 1.5 hr at 4°C. Supernatant was removed and 50 µl 1× NuPAGE LDS Sample Buffer was added to lyse pelleted virions. Virion samples were heated 72°C for 10 min, and 15 µl virion lysate was loaded into each well of a NuPAGE 4–12% Bis-Tris gels (ThermoFisher NP0329BOX). For cell lysates, 200 µl 1× NuPAGE LDS Sample Buffer (ThermoFisher NP0008) were added to each well, sonicated, and heated at 72°C for 10 min before loading into the NuPAGE 4–12% Bis-Tris gels. Blotted membranes were probed with mouse monoclonal anti-HIV-1 p24CA (183-H12-5C, NIH AIDS Reagent Program) and rabbit anti-HSP90 Polyclonal antibody (Proteintech 13171–1-AP).

## Virion unspliced viral gRNA extraction and quantification

HEK293T cells ($8 \times 10^5$) were transfected with 2 µg HIV-1 NL4-3 proviral constructs or CA mutants thereof in six-well plates. Cells were washed with PBS once the following day, and culture media was replaced with fresh media. After around 40 hr, culture supernatants were filtered through a 0.45 µm filter and digested with 4 U Turbo Dnase I (ThermoFisher AM2238) for 40 min at 37°C. Then 900 µl filtered virion supernatant was pelleted through a sucrose cushion and resuspended in 120 µl PBS. Viral RNA was extracted from 60 µl of resuspended virions using the NucleoSpin Virus Kit (Macherey-Nagel 740983.50). Viral RNA from each sample was diluted 30-fold, and 3 µl was used in RT-quantitative PCR (RT-qPCR) reactions (0.1 µl total for each reaction). RNA levels were determined with Power SYBR Green RNA-to-CT 1-Step Kit (ThermoFisher 4389986) using a StepOne Plus Real-Time PCR system (Applied Biosystems). Serial 10-fold dilutions of known copy numbers of HIV-1 NL4-3 plasmid was used to generate a standard curve for quantification. The RT-qPCR primers for amplification of unspliced viral gRNA were GAGCTAGAACGATTCGCAGTTA (forward) and CTGTCTGAAGGGATGGTTGTAG (reverse).

## Materials availability statement

All novel materials generated herein are available on request from the authors.

## Acknowledgements

We thank the staff of the Rockefeller University Genomics Resource Center for assistance and advice in the NGS sequencing of CLIP libraries. We thank Federico Comoglio, Ward Deboutte, Bert Vanmechelen, and Hugo Leonardo de Ávila (ORCID: 0000-0003-2739-003X) for their advice in CLIP data analysis. We thank Fengwen Zhang for the help in setting up the RT-qPCR assay. We thank members of the Bieniasz lab for helpful discussions. This work was supported by NIH grants R01AI50111 and U54 AI170660 to PDB. This article is subject to HHMI's Open Access to Publications policy. HHMI lab heads have previously granted a nonexclusive CC BY 4.0 license to the public and a sublicensable license to HHMI in their research articles. Pursuant to those licenses, the author-accepted manuscript of this article can be made freely available under a CC BY 4.0 license immediately upon publication.

## Additional information

### Funding

| Funder | Grant reference number | Author |
| --- | --- | --- |
| National Institute of Allergy and Infectious Diseases | U54 AI170660 | Paul D Bieniasz |
| National Institute of Allergy and Infectious Diseases | R01AI50111 | Paul D Bieniasz |
| Rockefeller University | | Xiao Lei |

The funders had no role in study design, data collection and interpretation, or the decision to submit the work for publication.

## Author contributions
Xiao Lei, Conceptualization, Formal analysis, Investigation, Methodology, Writing - original draft, Writing - review and editing; Daniel Gonçalves-Carneiro, Trinity M Zang, Investigation, Methodology; Paul D Bieniasz, Conceptualization, Supervision, Funding acquisition, Writing - original draft, Project administration, Writing - review and editing

## Author ORCIDs
Xiao Lei http://orcid.org/0000-0001-8641-7824
Daniel Gonçalves-Carneiro http://orcid.org/0000-0002-9333-1540
Paul D Bieniasz http://orcid.org/0000-0002-2368-3719

## Decision letter and Author response
Decision letter https://doi.org/10.7554/eLife.83548.sa1
Author response https://doi.org/10.7554/eLife.83548.sa2

## Additional files

### Supplementary files
• MDAR checklist

### Data availability
All data generated or analysed during this study are included in the manuscript and accompanying source data files.

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
