## [Editor Report]

This work presents valuable findings that advance our understanding of the roles of the CA domain in specific binding of HIV-1 Gag to the viral genomic RNA. The compelling evidence obtained using the modified CLIP-seq and chemical crosslinking approaches support the authors' conclusion that the initial Gag lattice formation mediated by CA is essential for Gag recognition of the 5' Ψ sequence. This work will be of interest to virologists working on gRNA packaging of not only HIV-1 but also other RNA viruses.

---

## [Decision Letter]

**Decision letter after peer review:**

Thank you for submitting your article "Initiation of HIV-1 Gag lattice assembly is required for cytoplasmic recognition of the viral genome packaging signal" for consideration by *eLife*. Your article has been reviewed by 3 peer reviewers, one of whom is a member of our Board of Reviewing Editors, and the evaluation has been overseen by Sara Sawyer as the Senior Editor. The following individual involved in review of your submission has agreed to reveal their identity: Julia C Kenyon (Reviewer #3).

Essential revisions:

1) Figure 2. Considering that Gag concentration within the cytosol may affect its binding kinetics, both with itself and with the RNA and that the mutations may affect the stability of the proteins, the expression levels of these proteins need to be measured.

2) Figure 4. Some CA mutants, such as E28A/E29A, Q219A and R229A, produced measurable amounts of virion particles, but their Psi binding ability was reduced. The RNA/Gag ratios of these mutants should be measured to address the concern whether the initial Gag binding to Psi dictates RNA packaging and thus strengthen the paper.

*Reviewer #1 (Recommendations for the authors):*

1. Figure 2B. Since the mutations may affect the stability of the proteins, the expression levels of these proteins should be measured. Different levels of the proteins may affect the binding specificity of the proteins to target RNA.

2. Figure 4. Some CA mutants, such as E28A/E29A, Q219A and R229A, produced measurable amounts of virion particles, but their Psi binding ability was reduced. Measuring the RNA/Gag ratios of these mutants would help to address the concern whether the initial Gag binding to Psi of these mutants dictates RNA packaging and thus strengthen the paper.

3. The manuscript should be double checked for typos (for example, "lost the ability to specific bind to Ψ" in the Summary).

*Reviewer #2 (Recommendations for the authors):*

1. Please provide a quantitative analysis of the crosslinking data shown in Figures 2C, 4A, and 5A (e.g., the band intensity of monomer, dimer, 3-6x, and 8/10x). Please see the comment #3 below.

2. Please provide a quantitative analysis of the virus particle production experiment shown in Figure 4 supplement 1. The Gag amounts in cell lysates vary substantially between Gag mutants, which may contribute to the difference in the virion release. In that case, the virion release defect observed with some CA mutants may be due to the Gag instability rather than disruption of CA-CA interfaces.

3. Figure 2C and Figure 4A. Chimeric Gag proteins, e.g., ccHex2-SP1-NC, formed up to hexamer, but CANC formed higher order multimers (i.e., 8x and 10x). Therefore, it is possible that the chimeric NC proteins failed to bind the Ψ element due to the lack of higher order multimers. Consistent with this possibility, some CANC mutants, which appear to this reviewer to show a defect in higher order multimerization but not dimer formation, e.g., R100A/S102A, T107A/T108A, and T110A/Q112A, fail to bind the Ψ element.

4. Discussion, second paragraph: The authors suggest that the basic building block of a Gag lattice is a Gag dimer, not a Gag hexamer. This is not consistent with the crosslinking data that appear to show no accumulation of Gag dimer when the CA trimer or hexamer interface is disrupted. It also contradicts the last sentence in the same paragraph.

5. Duchon et al. (2021) reported that in addition to the CA-CA dimer interface, the ability to bind membrane is necessary for a Gag molecule to recruit viral RNA to assembling particles (i.e., packaging). In this regard, it is interesting that the current study, which focuses on an earlier step, showed that specific recognition of viral RNA does not require membrane binding of Gag but still requires CA-CA interactions. It would be helpful for readers if the authors discuss the difference between this study and the study by Duchon et al. and suggest an integrated model about the sequence of the events.

6. Discussion, first sentence: The authors concluded that "the initiation of the assembly of the immature HIV-1 Gag lattice in the cytosol of infected cells is required for the specific initial interaction between Gag/CANC and the Ψ element of the gRNA". This statement may be misconstrued as suggesting that a small-scale lattice must pre-exist for the Gag-gRNA interaction. Strictly speaking, the data showed the need for the ability of Gag to form a Gag lattice, which may occur upon the initial (and otherwise transient) NC-Ψ element interaction.

---

## [Author Response]

Reviewer #1 (Recommendations for the authors):1. Figure 2B. Since the mutations may affect the stability of the proteins, the expression levels of these proteins should be measured. Different levels of the proteins may affect the binding specificity of the proteins to target RNA.

In the revised manuscript we’ve included quantitative western blot analyses of the expression levels of these proteins (Figure 2 Figure-supplement 2)

2. Figure 4. Some CA mutants, such as E28A/E29A, Q219A and R229A, produced measurable amounts of virion particles, but their Psi binding ability was reduced. Measuring the RNA/Gag ratios of these mutants would help to address the concern whether the initial Gag binding to Psi of these mutants dictates RNA packaging and thus strengthen the paper.

In the revised manuscript we measured the vRNA:Gag ratio for all the CA mutants that generated some level of extracellular particles (Figure 4 Figure-supplement 4). We were initially surprised to find that there were only minor variations in the vRNA:Gag ratios for these mutants. However, given that these mutants generated extracellular virions, they must, by definition, have been able to assemble. An essential step in assembly is the generation of an immature Gag lattice. Thus, these mutant full-length Gag proteins must have assembled an immature Gag lattice despite the fact that the CA-NC proteins do not appear to assemble into higher order multimers in the cytosol. We posit that the CA mutants that generate virions in the context of full length virus but do not generate high order CA-NC multimers in the cytosol harbour partial defects in immature lattice formation that are evident in the context of CA-NC in the cytosol, but are at least partly suppressed when Gag is targeted to membrane in the context of a full length Gag protein. This notion is consistent with a notion proposed by O’Caroll et al. (PMID 22993163), who invoke functional redundancy between membrane binding, CA-CA interaction and RNA binding in driving HIV-1 particle assembly. Notably, these results do not alter the interpretation that immature lattice assembly is required for psi binding, but do suggest that immature lattice assembly can occur either in the cytosol or at the plasma membrane in order to enable psi recognition. We have altered the text of the revised manuscript to reflect these ideas.

3. The manuscript should be double checked for typos (for example, "lost the ability to specific bind to Ψ" in the Summary).

We have checked the manuscript a thoroughly as we can for typos

Reviewer #2 (Recommendations for the authors):1. Please provide a quantitative analysis of the crosslinking data shown in Figures 2C, 4A, and 5A (e.g., the band intensity of monomer, dimer, 3-6x, and 8/10x). Please see the comment #3 below.

We have included densitometric line scans of these gels as figure supplements in the revised manuscript (See Figure 2—figure supplement 3, Figure 4—figure supplement 1, Figure 5—figure supplement 1)

2. Please provide a quantitative analysis of the virus particle production experiment shown in Figure 4 supplement 1. The Gag amounts in cell lysates vary substantially between Gag mutants, which may contribute to the difference in the virion release. In that case, the virion release defect observed with some CA mutants may be due to the Gag instability rather than disruption of CA-CA interfaces.

We have included this analysis in the revised manuscript (in the revised Figure 4—figure supplement 2.)

3. Figure 2C and Figure 4A. Chimeric Gag proteins, e.g., ccHex2-SP1-NC, formed up to hexamer, but CANC formed higher order multimers (i.e., 8x and 10x). Therefore, it is possible that the chimeric NC proteins failed to bind the Ψ element due to the lack of higher order multimers. Consistent with this possibility, some CANC mutants, which appear to this reviewer to show a defect in higher order multimerization but not dimer formation, e.g., R100A/S102A, T107A/T108A, and T110A/Q112A, fail to bind the Ψ element.

The three mutants mentioned by the reviewer clearly retain some ability to multimerize as assessed by the crosslinking assay, but the extent of multimerization is clearly reduced compared to WT CANC. We agree with the reviewer that the failure of the chimeric Gag proteins, such as ccHex2-SP1-NC, to bind Ψ could be due to their failure to assemble beyond hexamers, but it could also perhaps be due to a failure to precisely position the NC domains in a hexameric configuration that recognizes Ψ. We have added some sentences to the discussion to address this point.

4. Discussion, second paragraph: The authors suggest that the basic building block of a Gag lattice is a Gag dimer, not a Gag hexamer. This is not consistent with the crosslinking data that appear to show no accumulation of Gag dimer when the CA trimer or hexamer interface is disrupted. It also contradicts the last sentence in the same paragraph.

We agree with this interpretation – in fact the discussion paragraph in question is intended to present the evidence both for and against the notion that a dimer is a basic building block of the Gag lattice. We concede that this discussion in the original submission was confusing and clearly we failed to convey this adequately. We have modified and simplified the wording to be clearer in our view that the order of addition of Gag molecules to the to the lattice is unlikely to be via discrete dimers, trimers or hexamer intermediates and that all the interfaces contribute similarly to lattice assembly.

5. Duchon et al. (2021) reported that in addition to the CA-CA dimer interface, the ability to bind membrane is necessary for a Gag molecule to recruit viral RNA to assembling particles (i.e., packaging). In this regard, it is interesting that the current study, which focuses on an earlier step, showed that specific recognition of viral RNA does not require membrane binding of Gag but still requires CA-CA interactions. It would be helpful for readers if the authors discuss the difference between this study and the study by Duchon et al. and suggest an integrated model about the sequence of the events.

We have included a new paragraph in the discussion that includes a discussion of the Duchon et al. results. The new data in the revised manuscript that measures vRNA incorporation into virions generated by certain CA mutants (see above) and Figure 4—figure supplement 4 impinges on these issues and is discussed alongside the Duchon et al. findings. Overall, we think these data indicate that packaging can occur in the cytosol or at the plasma membrane, but that in either case assembly of a nascent lattice is required.

6. Discussion, first sentence: The authors concluded that "the initiation of the assembly of the immature HIV-1 Gag lattice in the cytosol of infected cells is required for the specific initial interaction between Gag/CANC and the Ψ element of the gRNA". This statement may be misconstrued as suggesting that a small-scale lattice must pre-exist for the Gag-gRNA interaction. Strictly speaking, the data showed the need for the ability of Gag to form a Gag lattice, which may occur upon the initial (and otherwise transient) NC-Ψ element interaction.

Conceded, we have changed the wording of this sentence to “we conclude that the initiation of the assembly of the immature HIV-1 Gag lattice in infected cells is required for the maintenance of the interaction between Gag/CANC and the Ψ element of the gRNA.”